# Superionic effect and anisotropic texture in Earth's inner core driven by geomagnetic field

Shichuan Sun [1,2], Yu He [1,2,3] ✉, Junyi Yang[1,2], Yufeng Lin [4], Jinfeng Li[4], Duck Young Kim [3], Heping Li[1,2] & Ho-kwang Mao [3]

Seismological observations suggest that Earth's inner core (IC) is heterogeneous and anisotropic. Increasing seismological observations make the understanding of the mineralogy and mechanism for the complex IC texture extremely challenging, and the driving force for the anisotropic texture remains unclear. Under IC conditions, hydrogen becomes highly diffusive like liquid in the hexagonal-close-packed (hcp) solid Fe lattice, which is known as the superionic state. Here, we reveal that H-ion diffusion in superionic Fe-H alloy is anisotropic with the lowest barrier energy along the c-axis. In the presence of an external electric field, the alignment of the Fe-H lattice with the c-axis pointing to the field direction is energetically favorable. Due to this effect, Fe-H alloys are aligned with the c-axis parallel to the equatorial plane by the diffusion of the north–south dipole geomagnetic field into the inner core. The aligned texture driven by the geomagnetic field presents significant seismic anisotropy, which explains the anisotropic seismic velocities in the IC, suggesting a strong coupling between the IC structure and geomagnetic field.

The complex and anisotropic Earth's inner core is revealed with an increase in seismological observations. It presents faster seismic velocity in the polar direction than in the equatorial direction[1–8], significant anisotropy changes with depth[9–15], and hemispherical dichotomy[15–20]. The mechanism for the anisotropic and heterogenous structure is critical for understanding the inner core history and its relationship with the outer core[21–23]. The lattice-preferred orientation (LPO) of Fe crystals has been proposed to explain the observed IC anisotropy[24–28]. The anisotropic velocity in hexagonal-close-packed (hcp) and body-centred-cubic (bcc) Fe can explain the difference in the polar and equatorial seismic velocities with the fast velocity axes parallel to Earth's rotation axis[24,26,27]. A complex model with different Fe phases combined with different orientations is required to account for the hemispherical anisotropy variations[28]. The aligned structure may be formed during the solidification

process[29], but the solidified anisotropic texture vanishes after an extended period of annealing[30]. Alternatively, anisotropy may be generated by the flow of Fe crystals, driven by differential growth[31,32] requiring a relatively high viscosity of more than $10^{18}$ Pa s. Maxwell stress has also been suggested as a driving force for anisotropic texture[33,34]. However, the driving forces of these mechanisms originate at the inner core boundary (ICB), making it difficult to understand the isotropic uppermost inner core (UIC)[17,35,36] and the formation of a highly anisotropic innermost inner core (IMIC)[12,13,37–40]. The UIC also presents quasi-hemispherical variations in velocities and attenuations, which are attributed to the asymmetric solidification/melting of Fe alloys at ICB[41,42]. Although different models have been developed to explain certain features of IC, the formation mechanism of heterogeneous and complex IC remains a mystery.

[1]Key Laboratory of High-Temperature and High-Pressure Study of the Earth's Interior, Institute of Geochemistry, Chinese Academy of Sciences, Guiyang 550081 Guizhou, China. [2]University of Chinese Academy of Sciences, 100049 Beijing, China. [3]Center for High Pressure Science and Technology Advanced Research, 201203 Shanghai, China. [4]Department of Earth and Space Sciences, Southern University of Science and Technology, 518055 Shenzhen, China. ✉e-mail: heyu@mail.gyig.ac.cn

Here, we provide a new mechanism to explain observed complex and anisotropic inner core structure based on the aligned superionic Fe-H crystals driven by dipole geomagnetic field.

## Results

### Seismic anisotropy in superionic Fe-H alloy

The superionic state is an intermediate state between solid and liquid and is suggested to exist in the interior of Earth and exoplanets[43–47]. In our recent study[43], we show that some light elements (H, O, and C) are stable at interstitial sites in hcp-Fe and diffuse like liquid in the solid Fe sublattice under IC conditions, indicating a superionic state. The superionic effect leads to elastic softening of these Fe alloys, resulting in seismic velocities close to the observations. Here, we further investigated the elasticity anisotropies of superionic Fe alloys by employing the ab initio molecular dynamics (AIMD) method and calculated the seismic wave velocities along different directions (Supplementary Discussion 1 and 2). We observed an unusual change in velocity anisotropy in superionic hcp-FeH$_{0.25}$ as the temperature increased from 0 to 6000 K (Fig. 1). When the temperature increases from 0 to 4000 K, the velocity along the c-axis gradually decreases, and the compressional velocity anisotropy (AV$_P$) diminishes to a minimum value of 3.9%. From 4000 to 6000 K, the a-axis becomes the fastest direction, and the anisotropy increases with temperature, reaching 5.3% at 6000 K. The slowest direction also exhibits small shifts to higher angles with increasing temperature. Beyond the temperature effect, we also evaluated velocity anisotropy in Fe-H alloys with different hydrogen content (Fig. 1b). FeH$_{0.0625}$ still presents the fastest direction along c-axis like pure hcp-Fe, which is consistent with pervious study[44]. Further increasing the hydrogen content leads to a reversal of the fastest direction. On the other hand, pressure only has a slight influence on velocity anisotropy. The fast direction reversal behavior, which is abnormal compared with other Fe alloys (Supplementary Discussion 3 and 4), may be caused by diffusive H-ions distributed at the interstitial sites within basal planes (discussed in Supplementary Discussion 2).

### H-ion diffusion anisotropy and preferred orientation in an external electric field

The migration of interstitial impurities in hcp alloys is mostly anisotropic[48,49]. The migration paths for the H-ion in hcp-FeH$_{0.25}$ are shown in Fig. 2a, and the barrier energies for migration along these paths at temperatures from 2000 to 6000 K are calculated using the climbing-image nudged elastic band (CINEB) method. The lattice parameters at different temperatures are obtained from hydrostatic AIMD simulations. The octahedral site (O-site) is a stable interstitial site for hydrogen, while the tetrahedral site (T-site) is metastable. Here, three hydrogen migration paths are considered, and the barrier energies are shown in Fig. 2. Paths 1 and 3 are direct migration from an O-site to a nearby O-site (O-O) along the a and c axes, respectively. Path 2 is indirect migration from an O-site to a nearby T-site and then to another O-site (O-T-O) along the a-axis direction. The barrier energies along the c-axis are the lowest, suggesting the most favorable path for H-ion migration. The barrier energies for the three paths mostly decrease with increasing temperature (Fig. 2c), and the anisotropic diffusion behavior is due to the non-ideal c/a ratio (discussed in Supplementary Discussion 5). We also calculated the diffusion activation enthalpies along different crystallographic orientations using AIMD simulations (Fig. 2d), and the activation enthalpy along the c-axis is the lowest, which is consistent with the CINEB result. Although the barrier energy is higher along the a-axis, the two equivalent migration paths along the a-axis (Supplementary Fig. 11) result in a high diffusion coefficient comparable to that along the c-axis at 6000 K. In this case, further investigation of the diffusion anisotropy in the presence of an electric field is necessary.

We used neural network potential (NNP)-based MD simulations adopting large supercells to study diffusion anisotropy in the presence of an external electric field. NNP, which was trained using the AIMD dataset, is applicable for quantum-accuracy MD simulations (Method & Supplementary Discussion 6). We conducted nonequilibrium molecular dynamics simulations (NEMD) on FeH$_{0.25}$ containing 640 atoms under external electric fields at 360 GPa and 6000 K using NNP. The external electric field promotes orientational H-ion diffusion along the field direction (Fig. 3a), and the internal energy of FeH$_{0.25}$ is lower when

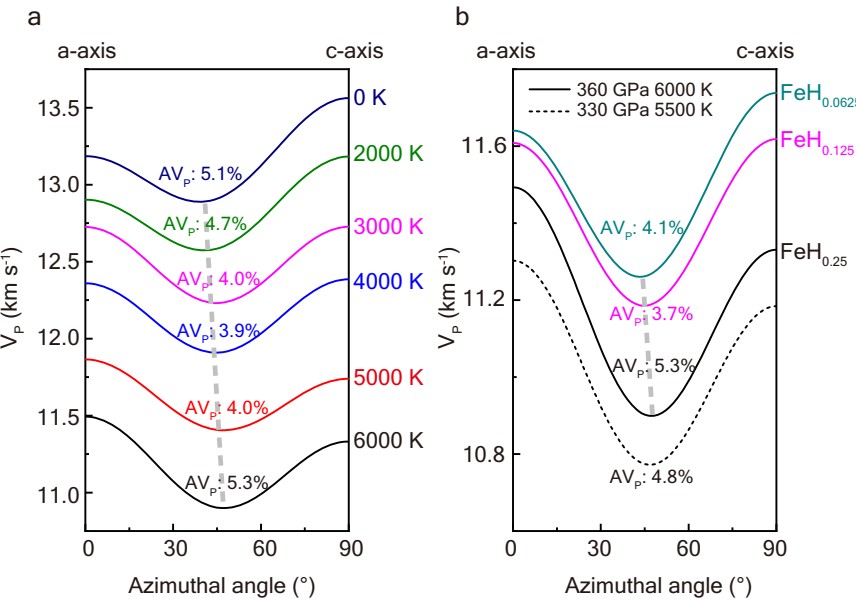

**Fig. 1 | Seismic wave velocity anisotropies in hcp-FeH$_{0.25}$. a** The velocity anisotropies (AV$_P$) in FeH$_{0.25}$ at 360 GPa and 0–6000 K. Different color curves represent velocities at temperatures from 0 to 6000 K, noting the percentage of the AV$_P$. The azimuthal angle of 0° corresponds to the velocity along the a-axis, while 90° corresponds to the c-axis. **b** The velocity anisotropies in Fe-H alloys with different H contents at 360 GPa and 6000 K. Different color curves represent velocities in FeH$_{0.25}$, FeH$_{0.125}$, and FeH$_{0.0625}$, and the velocities in FeH$_{0.25}$ at 330 GPa and 5500 K are shown with a dashed curve. The dashed gray lines show the trend of the slowest axis with increasing temperature and hydrogen content, respectively.

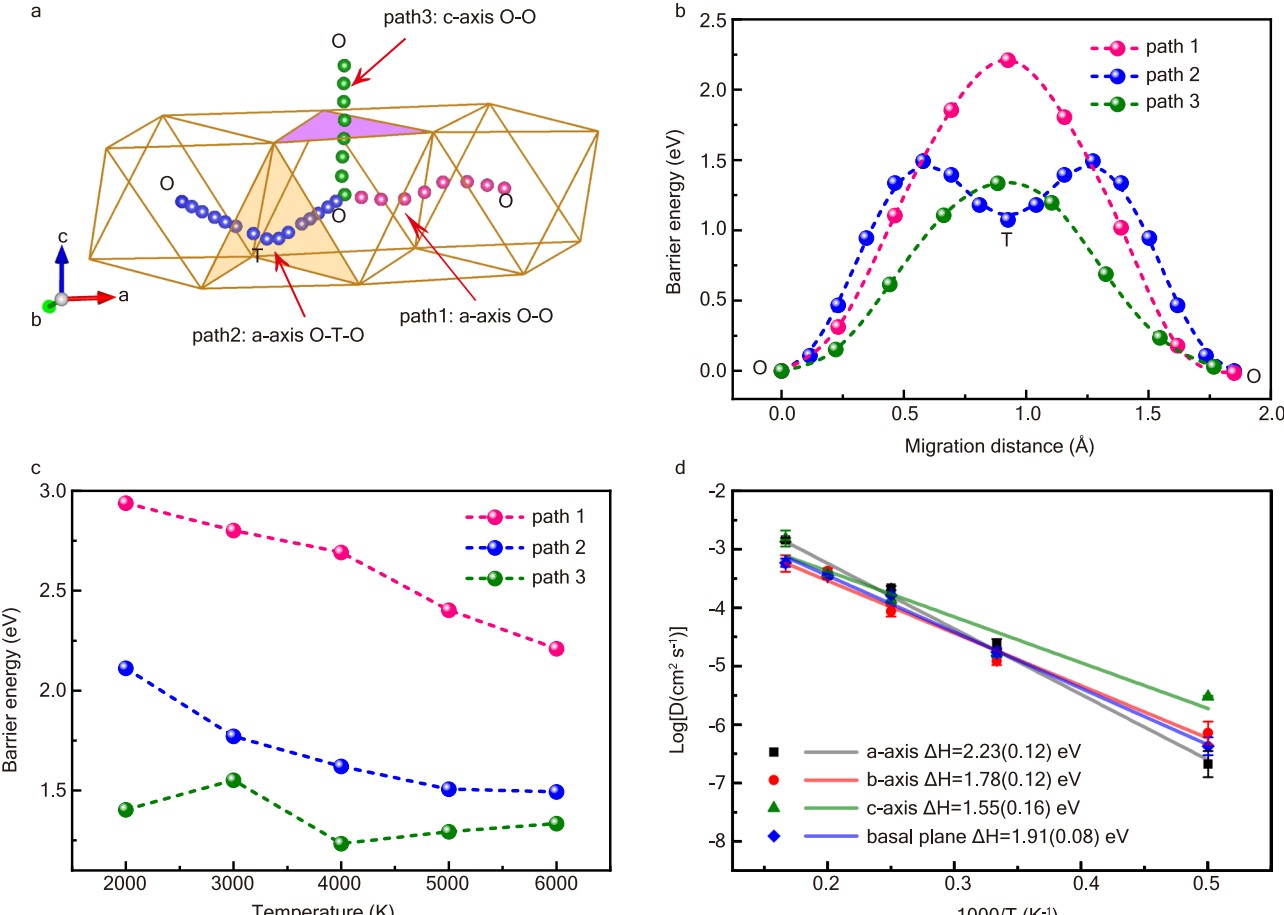

**Fig. 2 | Migration paths and barrier energies of H-ions in hcp-FeH$_{0.25}$ at 360 GPa and 2000–6000 K. a** The migration paths for H-ions in hcp-Fe. The octahedral and tetrahedral sites are noted with O and T. Pink and blue spheres represent O-O direct (path 1) and O-T-O indirect (path 2) paths in the basal plane. Green spheres represent the O-O path (path 3) along the c-axis. The framework of Fe is presented with blown lines. **b** The H-ion migration barrier energies as a function of migration distance at 360 GPa and 6000 K. **c** Barrier energies for different H-ion migration paths at 2000–6000 K. The barrier energies along path 1, path 2, and path 3 are shown with pink, blue, and green spheres. **d** H-ion diffusion coefficients (AIMD) in hcp-FeH$_{0.25}$ as a function of reciprocal temperature. The diffusion coefficients along the a-axis, b-axis, c-axis and basal plane at 360 GPa and 2000–6000 K are shown with black squares, red circles, green triangles, and blue diamonds. The data are linearly fitted, and the calculated activation enthalpies (ΔH) are noted. The error bars indicate the variation of linear fitting of mean square displacement (MSD) data.

the c-axis is parallel to the direction of the electric field (Fig. 3b, and c). In particular, the energy difference increases with simulation time; thus, even though the electric field intensity is much lower in the IC, a significant energy difference accumulates in the long term.

External electric fields drive orientational ionic flux in superionic materials, which generates diffusion-induced stress (DIS) in the lattice[50]. At temperatures above the recrystallization temperature, the DIS promotes recrystallization and the grain growth process, which minimizes the stress and internal energy in the lattice. The IC temperature is well above the recrystallization temperature of Fe alloys; thus, DIS drives the formation of oriented Fe-H crystals with the c-axes pointing toward the electric field direction.

Generally, we have shown several lines of evidence on the anisotropic H-ion diffusion behavior in superionic Fe-H. Superionic Fe-H alloys present both diffusion and seismic anisotropy under inner core conditions, which provides a new mechanism for the formation of anisotropic texture in the presence of anisotropic H-ion diffusion in external electric fields.

## Anisotropic structure driven by geomagnetic field

Dipole geomagnetic field may promote anisotropic H-ion diffusion in the IC. Thus, the distribution of the geomagnetic field in the IC is crucial to generate the anisotropic model for superionic Fe-H. The

observed geomagnetic field on Earth's surface represents only a fraction of the dynamo-generated field in the outer core and can only be extrapolated downwards to the CMB. The structure and intensity of the magnetic field in the core remain poorly constrained. Nevertheless, numerical geodynamo simulations have provided some insights into the core magnetic field[51]. The magnetic field at the ICB can be decomposed into poloidal ($B_P$) and toroidal ($B_T$) components[33,34,51]. The toroidal magnetic field lines are tangential to the ICB and may only diffuse into the very top region of the solid inner core due to the skin effect of the electromagnetic field. On the other hand, the poloidal magnetic field ($B_P$) can pass through the ICB and may penetrate into the entire inner core (Fig. 4a)[51]. The diffusion of the poloidal magnetic field leads to toroidal electrical currents according to Ampère's law ($J_T = \nabla \times B_P$). The electrical currents mainly consist of the azimuthal component, i.e., circulating along the lines of latitudes for a dipole-dominated magnetic field[51]. It is worth noting that here we did not consider the possible tilting of $B_P$. At the shallow depth of the IC (yellow region in Fig. 4a), the diffusion of H-ions is isotropic owing to the combined influence of $B_P$, $B_T$, and other possible mass fluxes driven by other mechanisms (e.g., thermal convection, viscous convection and concentration gradient) at the ICB, and no particular anisotropic texture can be formed at these depths, which explains the observed isotropic UIC layer. The averaged velocity of FeH$_{0.25}$ at 330 GPa and

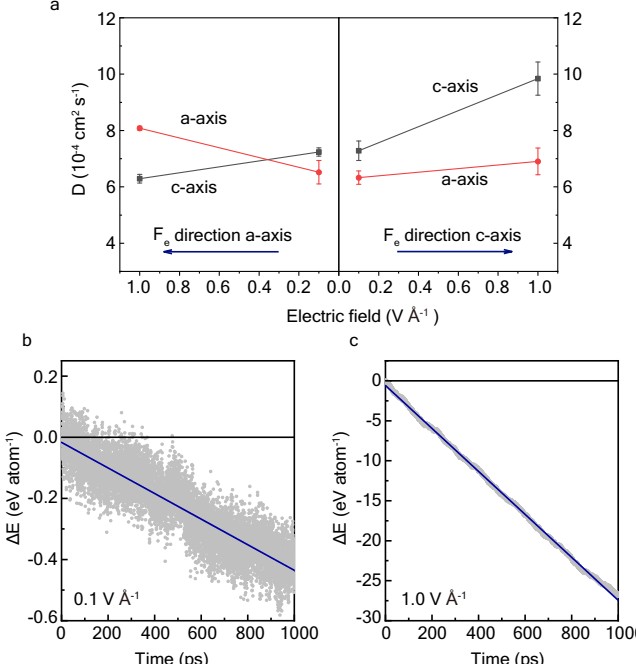

**Fig. 3 | H-ion diffusion anisotropy and energy differences in hcp-FeH$_{0.25}$ under external electric fields. a** H-ion diffusion anisotropy changes with the intensity and direction of the external electric fields. Diffusion coefficients along the a-axis and c-axis directions are shown with red and black symbols. The electric field ($F_e$) directions are shown with blue arrows. The error bars indicate the variation of linear fitting of MSD data. **b, c** Energy difference of FeH$_{0.25}$ ($\Delta E = E_{c\text{-axis}} - E_{a\text{-axis}}$) under the electric field of 0.1 V Å$^{-1}$ and 1.0 V Å$^{-1}$. The calculated energies are shown with gray dots, and the changes in $\Delta E$ with simulation time are linearly fitted with blue lines. The negative $\Delta E$ (below the black line) indicates that the c-axis of FeH$_{0.25}$ parallel to the external field direction is energetically favorable.

5500 K is 11.0 km s$^{-1}$, which fits the velocities of PREM[52] (Supplementary Fig. 2a).

The influence of B$_P$ becomes dominant with increasing depth in the blue region. The electric currents of B$_P$, circulating along latitudinal lines, can drive the LPO of the hcp-Fe-H alloy with the c-axis perpendicular to Earth's rotation axis. Here, two models comprising isotropic and anisotropic equatorial planes are considered (Supplementary Discussion 4). In the isotropic equatorial plane (IEP) model, c-axis of hcp-Fe-H crystals are aligned perpendicularly to the polar direction and randomly distributed in the equatorial plane; In this model, the change in the velocity as a function of angle ξ between the ray path and Earth's rotation axis is calculated by averaging the velocities of all the possible propagation direction paths corresponding to the different alignment of crystal with a rotatable c-axis in the equatorial plane (Supplementary Fig. 7a). The anisotropic equatorial plane (AEP) model is an ideal model. It assumes that an aligned pattern can form and allow the velocity variation with ξ to be consistent with the velocity change from the a-axis to the c-axis (Supplementary Fig. 7b). The velocity change along with ξ is calculated and compared with absolute and differential seismic travel-time data (Fig. 4b). The IC anisotropy is exhibited by taking the differential travel time of seismic phases in the core. More ultra-polar paths are included in the latest dataset (Fig. 4b)[15], and body waves travelling close to the polar side are approximately 2% faster than those propagating in the equatorial direction if anomalous positive South Sandwich Islands (SSI) data are excluded due to mantle heterogeneity[15,53]. In this case, the IEP model presenting a milder fluctuation of the average velocity at high angles fits the travel-time residuals better (PKPab-PKIKP), and

approximately 65%–80% of aligned FeH$_{0.25}$ is able to explain the observed seismic anisotropy (Supplementary Fig. 10).

In the IMIC (green region), the seismic anisotropy presents the slowest angle of ~45–50°, and the AEP model fits better with the absolute travel-time data[13] and PKPab-PKIKP data for IMIC[38] (Fig. 4c). This may suggest the presence of a highly anisotropic equatorial plane in the IMIC, which allows most seismic waves to travel along the c-axis in the equatorial plane. This possibility was also proposed based on observations of the coda-correlation wavefield[40], which may due to a different structure of the electric field in the IMIC generated by past geomagnetic field.

This mechanism establishes a connection between the inner core structure and electric field structure in the IC. It is most likely that the electromagnetic field in the IC is complex, as the dipole geomagnetic field is tilted and eccentric to Earth's rotational axis. Based on reconstructions of the magnetic field from the past 10,000 years, the geomagnetic field in the core can be described as eccentric to the west by ~100 km[54,55]. It may also suggest an eccentric electric field in the IC, and specific calculations on the electric field in the IC considering the characteristics of eccentricity and tilting are necessary. This may provide a clue to understanding hemispherical variations in velocity anisotropy and the complex IC structure.

Hydrogen is the most abundant element in solar systems. It may have been incorporated into the core during the early Earth stage due to the strong partition of hydrogen into liquid Fe over silicate melts under high pressure[56,57]. Here, we show that hcp-FeH$_{0.25}$ presents close density, seismic velocities, and velocity anisotropy comparable with observations of the IC. It is unsurprising that the simple Fe-H model cannot exactly match the seismological observations. In particular, Vs is still larger than that of the PREM data. As suggested by previous studies[58,59], Vs may be further reduced due to the presence of nickel and the effect of grain boundaries. The IC texture may be more complicated due to variations in temperature, pressure, composition, and possible light element concentration gradients. Intriguingly, superionic Fe-H alloys exhibit both seismic velocity and H-ion diffusion anisotropy, which establishes a connection between the anisotropic texture and magnetic field in the IC. The seismological anisotropy texture formed due to the anisotropic H-ion fluids in the geomagnetic field explains the polar-equatorial anisotropy and anisotropy variations with depth. Simulations of the magnetic field in the IC and additional seismological observations, such as coda correlation[40,60,61], will promote the understanding of the coupling between the IC structure and geomagnetic field and provide additional understanding of the complex texture and composition of the IC.

## Methods

### AIMD calculations on the seismic wave velocity anisotropy

The elastic properties and H-ion transportation properties were calculated using the Vienna Ab initio Simulation Package (VASP) based on density functional theory (DFT)[62,63]. We used the Perdew-Burke-Ernzerhof (PBE) exchange-correlation functional and projector augmented wave (PAW) pseudopotentials[64,65]. Under inner temperatures and pressures, the cell parameters of hcp-Fe-H alloys under hydrostatic conditions are calculated using ab initio molecular dynamics (AIMD) simulations within the canonical ensemble (constant N, V, T) with a time step of 1 fs. Supercells containing 64 Fe atoms (4 × 4 × 2) were used for AIMD simulations. Hydrogen atoms were randomly added to the octahedral interstitial sites in hcp-Fe to construct the Fe$_{64}$H$_{16}$ structure. The energy cut-off was 400 eV. These parameters are consistent with previous studies, and the convergence test suggests that a larger supercell and/or energy cut-off did not change our results[43]. A grid of simulations over different volumes and cell parameters was conducted for over 20,000 steps to obtain the hydrostatic

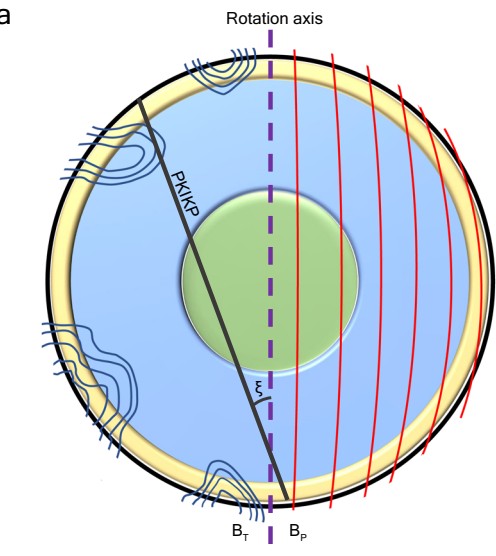

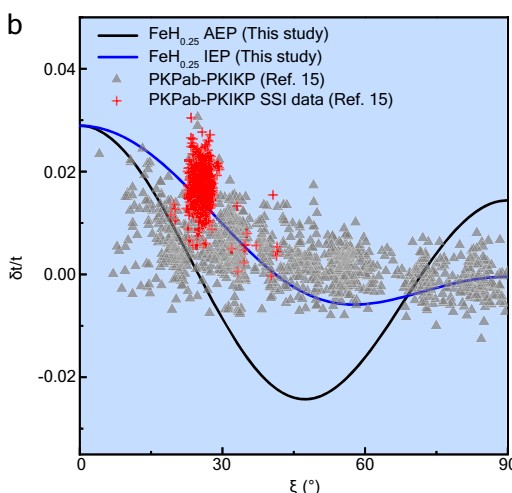

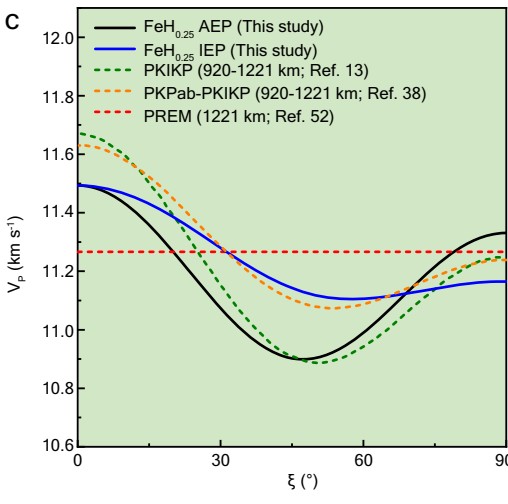

**Fig. 4 | Schematic diagram of the IC geomagnetic field and depth-dependent anisotropic texture change in comparison with the calculated velocity anisotropy in FeH$_{0.25}$. a** Poloidal (B$_P$) and toroidal (B$_T$) fields in the IC in the meridional plane. B$_P$ is shown with red curves in the right sphere, and B$_T$ is shown with blue curves in the left sphere. The dashed purple line is the rotation axis. The thick black line is the PKIKP ray path. PKIKP refers to the compressional wave path transmitted through the inner core. The angle between the ray path and Earth's rotation axis is ξ. The isotropic layer at the UIC is shown with a yellow region. The deeper layer presenting the slowest velocity in the equatorial direction is shown with the blue region. The IMIC with the slowest angle of ~ 45–50° from Earth's rotation axis is shown with a green region. **b** PKPab-PKIKP travel-time residuals[15] as a function of angle ξ are compared with the calculated compressional wave velocity anisotropies in the AEP (black curves) and IEP (blue curves) models. PKPab is the reference seismic wave phase traversing only the mantle and outer core. The data travelling from the South Sandwich Islands (SSI) to Alaska are shown with red crosses. **c** Calculated velocity anisotropy in FeH$_{0.25}$ in comparison with the velocities of PREM[52] (1221 km, red dashed line) and the anisotropic IMIC (920–1221 km, green and orange dashed curves) modes[13,38].

structure, the nonequivalent elastic constants $C_{11}$, $C_{12}$, $C_{13}$, $C_{33}$, and $C_{44}$ were calculated with the following distortion matrices:

$$\begin{pmatrix} 1+\delta & 0 & 0 \\ 0 & 1 & 0 \\ 0 & 0 & 1 \end{pmatrix}, \begin{pmatrix} 1 & 0 & 0 \\ 0 & 1 & \delta/2 \\ 0 & \delta/2 & 1 \end{pmatrix}, \text{and} \begin{pmatrix} 1 & 0 & 0 \\ 0 & 1 & 0 \\ 0 & 0 & 1+\delta \end{pmatrix}.$$

where $\delta$ is the magnitude of distortion. Different strains with $\delta$ equal to ±0.01, ±0.005, and 0 were added by:

$$a' = a(I+\varepsilon), \tag{2}$$

where a represents a 3 × 3 cell parameter matrix, ε represents added strain △ε, and I represents a 3 × 3 identity matrix. Thus, the elastic constants were calculated by solving the stress–strain relationship (Eq. 1).

We calculated the bulk modulus B and shear modulus G using the Voigt average scheme, which is proven to be more appropriate and accurate in calculating the seismic wave properties[66]. The compressional wave velocity $V_P$, shear wave velocity $V_S$, and bulk sound velocity $V_\Phi$ are calculated by:

$$V_P = \sqrt{\frac{B+\frac{4G}{3}}{\rho}}, V_S = \sqrt{\frac{G}{\rho}}, V_\Phi = \sqrt{\frac{B}{\rho}} \tag{3}$$

To further investigate the elastic anisotropy, we calculated the azimuthal angle-dependent velocity of superionic FeH$_{0.25}$ by solving the following Christoffel equation[67].

$$\rho V^2 = C_{ijkl}n_i w_j w_k n_l, (i,j,k,l=1,2,3), \tag{4}$$

where **n** is the propagation direction and **w** is the polarization direction. For hexagonal systems, we have:

$$\rho V_P^2 = C_{11} + (4C_{44}+2C_{13}-2C_{11})\cos^2\xi + (C_{33}+C_{11}-4C_{44}-2C_{13})\cos^4\xi, \tag{5}$$

The compressional wave velocity anisotropy $AV_P$ and the maximum shear wave splitting anisotropy $AV_S$ in Fe alloys (Supplementary Table 1) are calculated by:

$$AV_P = \frac{(V_P^{MAX}-V_P^{MIN})\times 200}{V_P^{MAX}+V_P^{MIN}}, \tag{6}$$

structure. The elastic constants were calculated by solving the stress–strain relations:

$$\sigma_{ij} = C_{ijkl}\varepsilon_{kl}, \tag{1}$$

where $\sigma_{ij}$ refers to the stress tensor, $\varepsilon_{kl}$ refers to the strain tensor, and $C_{ijkl}$ represents the fourth-order elastic modulus. For the hcp

$$AV_S = \left[ \frac{(V_{S1} - V_{S2}) \times 200}{V_{S1} + V_{S2}} \right]^{MAX}. \tag{7}$$

## Calculations on the H-ion transport anisotropy

The hydrostatic models adopted from AIMD simulations at 2000–6000 K were used for the migration barrier energy calculations. The barrier energies along different migration paths are calculated using the climbing-image nudged elastic band (CINEB) method[68]. This CINEB method duplicated a series of images (7 images in our calculations) between the starting point and the end point of migrating ions to simulate the intermediate states, with the positions of the starting point and the end point fixed. The actual diffusion pathway and migration barrier energy between the starting and end points are obtained by searching for the saddle point. Only the Γ point was adopted for k-point sampling to reduce the computational cost. The convergence check indicates that a denser k-mesh does not qualitatively affect our conclusion. The cut-off energy is 600 eV. To investigate the barrier energies of H-ion migration from one site to another, larger $8 \times 4 \times 2$ and $4 \times 4 \times 4$ supercells containing 160 atoms were used to calculate the barrier energies along the a-axis and c-axis, respectively. A detailed discussion is provided in Supplementary Discussion 5.

## Simulation on the H-ion diffusion in the presence of an external electric field

The software package DeepMD-kit[69] was used for the training of a neural network potential (NNP) for the superionic $FeH_{0.25}$ alloy at 360 GPa and 6000 K. The central idea in NNP is that the energy of a structure can be decomposed into energy contributions from constituent atoms,

$$E = \sum_i E_i (i = 1, 2, \ldots, N) \tag{8}$$

A neural network is used to fit the atomic energy, forces, and virial tensor with respect to the atomic coordinates, which describe the local environment of an atom. The dataset for NNP training contains 100,000 configurations, which are obtained from AIMD simulations in the canonical (NVT) ensemble at 360 GPa and 6000 K. The atomic coordinates, forces, total energy, and virial tensor of these structures are input to the training process to minimize the loss function

$$L\left(p_\varepsilon, p_f, p_\xi\right) = \frac{p_\varepsilon}{N} \triangle E^2 + \frac{p_f}{3N} \sum_i |\triangle F_i|^2 + \frac{p_\xi}{9N} ||\triangle \Xi||^2 \tag{9}$$

where $\triangle E$, $\triangle F_i$, and $\triangle \Xi$ denote the root mean square (RMS) error in energy, force, and virial, respectively. $p_\varepsilon, p_f, p_\xi$ are prefactors. The embedding and fitting net sizes are (25, 50, 100) and (240, 240, 240), respectively. The radial cut-off was set to 9.5 Å, with smoothing starting from 0.5 Å. The training steps were tested and set to 1,000,000 with a decay step of 5000. The root mean square errors are 4 meV atom$^{-1}$ for the energies, 256 meV Å$^{-1}$ for the forces, and 24 meV atom$^{-1}$ for the virial stresses on the testing set at 360 GPa and 6000 K (Supplementary Fig. 12). A detailed discussion of NNP training in this study is shown in Supplementary Discussion 6.

To further investigate the anisotropic diffusion behavior of $FeH_{0.25}$ under external electric fields, we performed nonequilibrium molecular dynamics simulations (NEMD) with machine learning NNP using the large-scale atomic/molecular massively parallel simulator (LAMMPS) package[70] within the NVT ensemble (T = 6000 K, P = 360 GPa). In this work, we expanded the structure of $Fe_{64}H_{16}$ using AIMD simulations into a $2 \times 2 \times 2$ supercell containing 640 atoms. The

simulated system is subject to periodic boundary conditions in all spatial directions. The temperature is controlled at 6000 K via a Nośe-Hoover thermostat and barostat. All simulations are performed for 1000 ps with a time step of 1 fs. Uniform static electric fields with intensities of 0.1 and 1.0 V Å$^{-1}$ were set along the OX-axis (the a-axis of hcp-$FeH_{0.25}$) and the OZ-axis (the c-axis of hcp-$FeH_{0.25}$), respectively. The incorporation of an electric field into MD simulations can be achieved by adding a force $\mathbf{F} = q\mathbf{E}$ with a charge of q. The atomic charges of Fe and H were set to +0.075 e and −0.3 e according to the Bader charge calculation (Supplementary Table 3). The total energy in MD simulations was calculated by counting the contributions from potential energy, kinetic energy and electric-field-induced energy change. The detailed simulation results are provided in Supplementary Discussion 6.

## Data availability

All data are available in the paper or in the supplementary materials. The raw data are available from the 4TU Center for Research Data (https://313data.4tu.nl/articles/dataset/Data_underlying_the_survey_on_Elasticity_of_hcp_Fe-314H_alloy_/17121380) Source data are provided with this paper.

## Code availability

The Vienna Ab initio Simulation Package is proprietary software available for purchase at https://www.vasp.at/. Lammps software is available at https://www.lammps.org/. Deepmd-kit code is available at https://github.com/deepmodeling/deepmd-kit.

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

## Acknowledgements

The seismic data in comparison with our models are obtained from references Dziewonski & Anderson 1981; Ishii & Dziewonski 2002; Sun & Song 2008; and Brett & Deuss 2020. We acknowledge H. Brett and A. Deuss for providing us with the original seismic data of their publication. We also acknowledge X. Sun and X. Song for data transformation. We acknowledge the support of the National Natural Science Foundation of China (42074104, U1930401), CAS Youth Interdisciplinary Team (JCTD-2022-16), and Youth Innovation Promotion Association of CAS (2020394). This study was also supported by the Strategic Priority Research Program (B) of the Chinese Academy of Sciences (XDB 18010401) and Guizhou Provincial 2020 Science and Technology Subsidies (No. GZ2020SIG). Numerical computations were performed at the Hefei Advanced Computing Center, Shanghai Supercomputer Center, and National Supercomputer Center in Guangzhou. The computing resource of this work was also provided by the Bohrium Cloud Platform (https://bohrium.dp.tech), which is supported by DP Technology.

## Author contributions

Y.H. conceived and designed this work. Y.H. carried out diffusion property calculations. S.S. carried out calculations on seismic velocities. S.S. and J. Y. trained the NNP and conducted MD simulations under electric fields. Y. L. and J. L. analyzed the electric current in the inner core and wrote the related discussion. Y.H. and S.S. conducted the data analysis and wrote the original draft. Y.H. S.S. D.Y.K., H. L. and H.-k. M. discussed the geophysical implications. All the authors contributed to the review and editing.

## Competing interests

The authors declare no competing interests.
