## [Peer Review File · Nature Communications]

Superionic effect and anisotropic texture in Earth's inner core driven by geomagnetic fieldREVIEWER COMMENTS

Reviewer #1 (Remarks to the Author):

The paper presents a study on a solid iron-hydrogen alloy using first principles methods based on density functional theory

The main findings of the paper are velocity anisotropies in different crystal directions and the suggestion of superionicity

behaviour of hydrogen, which would diffuse at different rates in different crystal directions. These differences are

suggested on the basis of energy barriers for diffusion, which are calculated using the climbing image nudged elastic band (CINEB) method.

The paper is interesting and in my view and puts forward a valuable suggestion to explain the seismic anisotropy which

sees seismic waves traveling faster in the polar than in the equatorial direction.

However, at present I believe the method used to compute hydrogen diffusion rates in different crystallographic directions

is misleading. The authors present results for temperatures between 2000 and 6000 K, but the CINEB method is a zero

temperature method. At finite temperature entropic effects play a role, but the CINEB method only relies on energies,

ignoring these entropic effects altogether. It would seem that the authors include the effect of temperature only through

thermal expansion. Given the (zero T) energy barriers, at 6000 K hydrogen should diffuse pretty quickly in the crystal,

and therefore perhaps molecular simulations would be able to directly calculate diffusion coefficients.

Such a calculation would

automatically include energy and entropy effects, and so it would be much more convincing if the conclusions of the paper

were based on such simulations.

Reviewer #2 (Remarks to the Author):

Review

Xiaodong Song, Peking U

The authors presented a mechanism for the IC anisotropy under the superionic effect, which is revealed by their recent paper (Yu et al., 2022 Nature). The prominent anisotropy of the IC in seismic velocities and characterization of its great complexities have been a major breakthrough in the studies of the Earth's deep interior. However, so far, the convincing mechanism for the anisotropy and its patterns has been lacking. This study offers a brand-new mechanism. I find the idea exciting and the analyses quite impressive. The topic is also of great interest to the communities of seismology and geodynamics, besides mineral physics. I recommend its publication but with a major revision.

1. Story line.

I can understand the general story line. But the presentation can (needs to) be clearer. From my understanding, it goes like below.

(1) Section on seismic anisotropy in superionic Fe-H alloy and sections S1-S4.

The study shows that FeH alloy is anisotropic in seismic velocity and can fit seismic data too. This is

good. However, this shows just the possibility. Previous studies have shown elastic anisotropy of Fe crystals (hcp, bcc), which can also fit seismic observations.

(2) Section on H-ion diffusion anisotropy and preferred orientation in external electric field and sections S5-6. The arguments provide a new physical mechanism for elastic anisotropy. This is very interesting, but obviously it needs to be evaluated by an expert on the subject.

(3) Section on anisotropic structure driven by geomagnetic field. This goes a step further to link the physical mechanism to real earth. Combining with 1) and 2) above, it provides possible consistent mechanism for the IC anisotropy. The model tries to explain the depth dependence of the anisotropy from the ICB to the IMIC. This is quite exciting. Linkage to geomag is important: it is readily available and can readily explain the polar anisotropy.

On the other hand, the linkage with the geomagnetic field is still qualitative at the moment.

Clarification on the presentation. Consider adding introductory sentences on each section (in main text or supplement) and adding summary sentences. This may provide a better guide to readers.

A separate example is S7. A simple sentence can be added at the start.

2. Early studies of inner core anisotropy structure.

In general the authors were diligent in tracking down studies of IC anisotropy and cited many references, but some important early contributions are missing in various places (Please go over all the references, starting from the first sentences of the paper).

Poupinet et al. 1981 was first to observe inner core anomalies. Morelli et al. 1986 and Woodhouse et al. 1986 were first to propose the idea of IC anisotropy. It was debated for many years, but generally accepted after the confirmation studies:

Creager 1992

Song and Helmberger 1993

Tromp 1993

Early studies of the depth dependence:

Shearer 1994 JGR

Song and Helmberger 1995 JGR

Initial studies on hemispherical variations:

IC anisotropy: Tanaka and Hamaguchi 1997

Isotropic velocity (topmost): Niu and Wen 2001 nature

Proposal of UIC/LIC transition structure:

Song and Helmberger 1998 Science

IMIC:

Initial proposal: Ishii and Dziewonski 2002 PNAS

Confirmation and characterization: Sun and Song 2008 EPSL

Equatorial anisotropy: Wang and Song 2015 NGeo

First model of 3D (2.5D) anisotropy:

Sun and Song 2008 PEPI

3.

Description of the IEP and AEP models needs to be clarified. L138-141, AEP model, is not clear. c-axis is at a fixed direction(s) (not random)? Description of Fig. S7 is unclear. Many symbols are not described. The orange arrows in the top panel and the pink symbols in the lower panel are confusing (and the figure is supposed to be 3D, making it even harder to image).

4. Like above, descriptions of all figure captions need to be carefully revised. In Figure 3, it's hard to find description of "b" and "c".

5. UIC. The mechanism for isotropic UIC is quite terse (L122-125). You need how it works. Also, what "possible mass fluxes at the ICB"? How do they affect anisotropy of uppermost 50 km or so?

6. IMIC. The model by Ishii and Dziewoński (2002) is based on ISC travel times (not so accurate) and a bit outdated. There are various models with updated results of IMIC radius and anisotropy pattern (e.g., Sun and Song 2008 EPSL; Ref. 48 Wang et al. 2015 Ngeo; Stephenson et al., 2020 JGR). Can your model fit well with more recent results?

A more interesting question is: Can your model explain the equatorial anisotropy in the IMIC, such as proposed by Wang et al. 2015? The key observation was a symmetry axis nearly parallel to the equatorial plane, rather than the change of the form of the anisotropy and how the anisotropy changes with ray angles.

7. Anisotropy complexities.

The authors also attempt at some IC anisotropy complexities, such as hemispherical dichotomy (L169-171). However, it is qualitative (which is fine at this stage), but the explanation for the hemispherical variation is not convincing. The "hemispherical" pattern is not exactly hemispherical (Sun and Song 2008 PEPI) either. In general, the anisotropy is quite complex, is fully 3D. On the other hand, it's likely that the geomagnetic field and electric current in the IC are also likely complex. That seems much more convincing cause for the anisotropy complexities. With field from geodynamo simulations, quantitative arguments can certainly be worked out.

8. Time varying anisotropy?

Magnetic field certainly changes with time. As it changes, will the IC anisotropy changes too? How long does it take? Temporal changes of the IC have been well observed (Song and Richards 1996 Nature ...). Would the anisotropy change be observable in human time scale?

9. Anisotropy models (IEP and AEP) (again).

The authors consider that the texture alignment of the Fe-H alloy is driven by the geomagnetic field. The angle ξ defined in this study is based on the Earth's rotation axis, but the geomagnetic pole is not the same as the rotation axis. This should be explained or at least pointed out in the manuscript or supplement.

Minor comments:

L57. "these alloys" -> clarify. "these alloys (with the light elements)"?

L62. "anisotropic compressional velocity (AVp)". This is confusing. Do you mean "compressional velocity anisotropy"?

L105-108. "Generally, we have .. ". This seems a summary sentence of the section. You probably should put it in a separate paragraph.

L391 (Figure 1b), the thick dashed gray line connects the troughs of the top three curves, but not including the bottom one. A mistake or on purpose? It looks a bit weird.

L420, Figure 4. You may illustrate the definition of ξ in (a), instead of in the lengthy caption, by adding a raypath and labeling the angle in a suitable place.

In plots of dt/t , such as Figure 4b, tell how the curves are calculated. At what distance/sampling depth?

Reviewer #3 (Remarks to the Author):

Sun et al., conducted theoretical calculations on elastic properties of the superionic Fe-H alloys at the inner core relevant P-T as well as electric field conditions. Their results showed that the geomagnetic field of inner core will align the c-axis of superionic Fe-H parallel to the equatorial plane in deeper depth instead of the uppermost region (both B_p and B_t exist). This can well explain the observed depth-dependent anisotropy in the inner core. Although the reason for the hemispheric dichotomy is still unclear, this study provides a new aspect to understand the possible coupling and interaction between the inner core materials with the Earth's geomagnetic field. There are some questions after reading the manuscript.

The authors calculated the AVp ranging 3.9-5.3% up to 6000 K, which is the maximum anisotropy of the hcp superionic Fe-H alloy. From Figures 1. S6 and S8, I found that the velocity difference between c- and a-axis is relatively small, about 1.7% at 6000 K and almost 0% at 4000 K. Under geomagnetic field, the c-axis of superionic Fe-H is aligned to be perpendicular to the Earth's rotation axis and the a-b plane is parallel. I wonder whether the velocity difference in c and a-axis can well explain the seismically-observed anisotropy magnitude in the inner core.

The authors found the the velocity of superionic Fe-H in c-axis becomes smaller than that in a-axis when temperature increasing to 5000 and 6000 K (so-called reversal of fastest direction). Such a phenomenon is not observed by Wang et al., calculating Fe₆₄H₄ at 6500 K and 360 GPa. What is the reason? In He et al., 2022, they suggested that Fe-O and Fe-C alloys can also be superionic phases at inner core relevant P-T. I wonder if the C and O would change the story or the observation here.

Based on Figure 4b, the anisotropy in the inner core can be roughly divided to three layers, the uppermost (almost no anisotropy), the middle (some anisotropy), and the innermost (strong anisotropy) regions. It is conceivable that the existence of both B_p and B_t in the uppermost region make it almost no anisotropy. However, it is unclear to me why there is a great increase in anisotropy in the innermost region from the manuscript. At lines 150-151, the authors mentions "This may suggest the presence of a highly anisotropic equatorial plane in the IMIC". I wonder what is such a plane and why it can exist? Can it be related to the properties of superionic Fe-H? The depth starting from? Please clarify or elaborate.

Although the existence of the superionic Fe-H can explain the anisotropy with depth to some extent, it still unclear how it causes the hemispheric dichotomy. Some efforts or attempts may be worthwhile on this part.

Minor comments:

In Figure 4, the caption should be "c". There is no "d".

In Methods, the AVs is actually the maximum shear wave splitting anisotropy instead of the shear wave anisotropy.

In figure S2, P, T, and H contents are three variables that change. It is unclear from the caption the exact fixed values in each category.

In tables S1 and S2, the c_{ij} has a unit of GPa?

Response Letter

Response to Reviewer #1:

The paper presents a study on a solid iron-hydrogen alloy using first principles methods based on density functional theory. The main findings of the paper are velocity anisotropies in different crystal directions and the suggestion of superionic behaviour of hydrogen, which would diffuse at different rates in different crystal directions. These differences are suggested on the basis of energy barriers for diffusion, which are calculated using the climbing image nudged elastic band (CINEB) method.

The paper is interesting and in my view puts forward a valuable suggestion to explain the seismic anisotropy which sees seismic waves traveling faster in the polar than in the equatorial direction. However, at present I believe the method used to compute hydrogen diffusion rates in different crystallographic directions is misleading.

[Authors]: Thank you for your positive and constructive comments. We have improved our manuscript according to your valuable suggestions.

The authors present results for temperatures between 2000 and 6000 K, but the CINEB method is a zero temperature method. At finite temperature entropic effects play a role, but the CINEB method only relies on energies, ignoring these entropic effects altogether. It would seem that the authors include the effect of temperature only through thermal expansion. Given the (zero T) energy barriers, at 6000 K hydrogen should diffuse pretty quickly in the crystal, and therefore perhaps molecular simulations would be able to directly calculate diffusion coefficients. Such a calculation would automatically include energy and entropy effects, and so it would be much more convincing if the conclusions of the paper were based on such simulations.

[Authors]: Thank you. It is a very important question. We already performed such calculations using the AIMD method, as provided in the supplementary information (SI,

Fig. S12). The fitted activated enthalpies are consistent with CINEB calculations. The H-ion diffusion coefficients along the a-axis and c-axis are almost identical at 6000 K, although the migration barrier energy along the c-axis is higher. This is because there are two equivalent migration paths for H ions along the a-axis (discussed in SI & Fig. S11). This is why we carried out MD simulations in the presence of external electric fields. The results are shown in Fig. 3. The results support our idea that hydrogen diffusion along the c-axis is energetically favorable. These results provide solid evidence for the anisotropic H-ion diffusion behavior in superionic Fe-H. To avoid further misunderstandings, we replotted Fig. 2, and the following sentences were added to the main text.

Lines 92-95: "...Although the barrier energy is higher along a-axis, the two equivalent migration paths along a-axis (Supplementary Fig. 11) result in a high diffusion coefficient comparable to that along c-axis at 6000 K. In this case, further investigation on the diffusion anisotropy in the presence of electric field is necessary."

Lines 97-98: "We used neural network potential (NNP)-based MD simulations adopting large supercells to study the diffusion anisotropy in the presence of an external electric field. NNP..."

Fig. 2 | Calculated migration paths and barrier energies of H-ions in hcp-FeH_{0.25} at 360 GPa and 6000 K. **(a)** The migration paths for H-ions in hcp-Fe. The octahedral and tetrahedral sites are noted with O and T. Pink and blue spheres represent O-O direct (path 1) and O-T-O indirect (path 2) paths in the basal plane. Green spheres represent the O-O path (path 3) along the c-axis. The framework of Fe is presented with blown lines. **(b)** The H-ion migration barrier energies as a function of migration distance at 360 GPa and 6000 K. **(c)** Barrier energies for different H-ion migration paths at 2000-6000 K. The barrier energies along the path 1, path 2, and path 3 directions are shown with pink, blue, and green spheres. **(d)** Calculated H-ion diffusion coefficients (AIMD) in hcp-FeH_{0.25} as a function of reciprocal temperature. The diffusion coefficients along the a-axis, b-axis, c-axis and basal plane at 360 GPa and 2000-6000 K are shown with black squares, red circles, green triangles, and blue diamonds. The data are linearly fitted, and the calculated activation enthalpies are noted.

Response to Reviewer #2:

The authors presented a mechanism for the IC anisotropy under the superionic effect, which is revealed by their recent paper (Yu et al., 2022 Nature). The prominent anisotropy of the IC in seismic velocities and characterization of its great complexities have been a major breakthrough in the studies of the Earth's deep interior. However, so far, the convincing mechanism for the anisotropy and its patterns has been lacking. This study offers a brand-new mechanism. I find the idea exciting and the analyses quite impressive. The topic is also of great interest to the communities of seismology and geodynamics, besides mineral physics. I recommend its publication but with a major revision.

[Authors]: We are grateful for the recognition of our study. We revised the manuscript according to your suggestions.

1. “Story line. I can understand the general story line. But the presentation can (needs to) be clearer. From my understanding, it goes like below.

(1) Section on seismic anisotropy in superionic Fe-H alloy and sections S1-S4.

The study shows that FeH alloy is anisotropic in seismic velocity and can fit seismic data too. This is good. However, this shows just the possibility. Previous studies have shown elastic anisotropy of Fe crystals (hcp, bcc), which can also fit seismic observations.

[Authors]: Thank you for the comment. Previous studies focused on the anisotropy of pure Fe. Here, we investigated the effect of light element hydrogen and superionicity. In this study, we show that seismic anisotropy, absolute velocities and densities all fit with seismic data in our model.

(2) Section on H-ion diffusion anisotropy and preferred orientation in external electric field and sections S5-6. The arguments provide a new physical mechanism for elastic anisotropy. This is very interesting, but obviously it needs to be evaluated by an expert

on the subject.

[Authors]: Thank you for the positive comment.

(3) Section on anisotropic structure driven by geomagnetic field. This goes a step further to link the physical mechanism to real earth. Combining with 1) and 2) above, it provides possible consistent mechanism for the IC anisotropy. The model tries to explain the depth dependence of the anisotropy from the ICB to the IMIC. This is quite exciting. Linkage to geomag is important: it is readily available and can readily explain the polar anisotropy.

[Authors]: Thank you for the positive comment.

On the other hand, the linkage with the geomagnetic field is still qualitative at the moment.

[Authors]: We agree with the reviewer that more studies in cooperation with experts in seismology and geodynamo simulation are needed to construct a detailed geomagnetic field-driven inner core model, which may provide more details and explain the complex inner core structure.

Clarification on the presentation. Consider adding introductory sentences on each section (in main text or supplement) and adding summary sentences. This may provide a better guide to readers. A separate example is S7. A simple sentence can be added at the start.”

[Authors]: Thank you for your valuable suggestions. We added/improved the following summary sentences in the revision:

In the main text:

Lines 61-63: “Here, we further investigated the elasticity anisotropies of superionic Fe alloys by employing the ab initio molecular dynamics (AIMD) method and calculated the seismic wave velocities along different directions.”

Lines 97-98: “We used neural network potential (NNP)-based MD simulations adopting large supercells to study the diffusion anisotropy in the presence of an external electric field.”

Lines 115-118: “Generally, we have shown several lines of evidence on the anisotropic H-ion diffusion behavior in superionic Fe-H. Superionic Fe-H alloys present both diffusion and elastic anisotropy under inner core conditions, which provides a new mechanism for the formation of anisotropic texture in the presence of anisotropic H-ion diffusion in external electric fields.”

Line 121-123: “Dipole geomagnetic may promote anisotropic H-ion diffusion in the IC. Thus, the distribution of the geomagnetic field in the IC is crucial to generate the anisotropic model for superionic Fe-H. ”

In the supplementary information:

Lines 52-53: “In this section, we provide a detailed calculation method and data for the elasticity and seismic velocities of superionic Fe-H alloys.”

Line 140-141: “In this section, we provide the detailed method for seismic velocity anisotropy and the discussion on the relationship between seismic anisotropy and lattice c/a ratio.”

Line 191: “Here, elastic anisotropies in Fe alloys are compared and discussed.”

Line 210: “In this section, the velocity anisotropies are compared with observations.”

Line 304-305: “In this section, we provide the detailed methods and discussions for diffusion properties of Fe-H alloys.”

2. “Early studies of inner core anisotropy structure. In general, the authors were diligent in tracking down studies of IC anisotropy and cited many references, but some important early contributions are missing in various places (Please go over all the references, starting from the first sentences of the paper).”

Poupinet et al. 1981 was first to observe inner core anomalies. Morelli et al. 1986 and Woodhouse et al. 1986 were first to propose the idea of IC anisotropy. It was debated for many years, but generally accepted after the confirmation studies:

Creager 1992

Song and Helmberger 1993

Tromp 1993

Early studies of the depth dependence:

Shearer 1994 JGR

Song and Helmberger 1995 JGR

Initial studies on hemispherical variations:

IC anisotropy: Tanaka and Hamaguchi 1997

Isotropic velocity (topmost): Niu and Wen 2001 nature

Proposal of UIC/LIC transition structure:

Song and Helmberger 1998 Science

IMIC:

Initial proposal: Ishii and Dziewonski 2002 PNAS

Confirmation and characterization: Sun and Song 2008 EPSL

Equatorial anisotropy: Wang and Song 2015 NGEA

First model of 3D (2.5D) anisotropy:

Sun and Song 2008 PEPI

[Authors]: Thank you for your valuable suggestion and detailed guidelines for the references. We did learn a lot from them. We reorganized our citations and included these crucial works in the revised manuscript.

3. “Description of the IEP and AEP models needs to be clarified. L138-141, AEP model, is not clear. c-axis is at a fixed direction(s) (not random)? Description of Fig. S7 is unclear. Many symbols are not described. The orange arrows in the top panel and the pink symbols in the lower panel are confusing (and the figure is supposed to be 3D, making it even harder to image).”

[Authors]: Thank you for this important question. The AEP model considers the anisotropic alignment of the c-axis in equatorial planes generated by the electric field in the equatorial direction. However, we cannot construct the anisotropic structure at the moment because there are few studies on the electric field in the inner core. Thus, we considered the most extreme condition. The AEP model assumes that the velocity variation with ξ is consistent with the velocity change from the a-axis to the c-axis (we assume the pattern is well aligned to fit this condition). It is an ideal model, but it provides additional information when it is compared with the seismic anisotropy in IMIC. This model suggests that the IMIC presents strong equatorial anisotropy, which was also proposed based on observations of the coda-correlation wavefield (Wang et al. Nat. Geo. 2015). It is interesting but we need to know the distribution of the electric field before we can construct possible equatorial patterns and compare them with seismological observations. We also revised Fig. S7 to make it clearer.

Supplementary Fig. 7 | Anisotropic models for hcp-FeH_{0.25} with the basal plane along Earth's rotation axis. (a) Isotropic equatorial model (IEP) model with the c-axis randomly distributed in the equatorial plane (inset). This model is viewed as a cylindrically averaged aggregate with a basal plane parallel to the rotation axis (black circular arrow). (b) The c-axis is aligned in some patterns (a possible pattern is shown in the inset) to allow the velocity variation with ξ to be consistent with the velocity change from the basal plane to the c-axis (black arrows).

4. “Like above, descriptions of all figure captions need to be carefully revised. In Figure 3, it’s hard to find description of “b” and “c”.”

[Authors]: Thank you for your careful reading and kind suggestion. We have rechecked our figure captions and corrected the mistakes in the revised manuscript.

5. “UIC. The mechanism for isotropic UIC is quite terse (L122-125). You need how it works. Also, what “possible mass fluxes at the ICB”? How do they affect anisotropy of uppermost 50 km or so?”

[Authors]: Thank you for the question. Based on this mechanism, the diffusion should be random in the influence of B_P and B_T . At UIC, the temperature is quite close to the melting temperature of the inner core material, thus, the mass fluxes may be driven by thermal convection, viscous convection and even the concentration gradient of light elements. Here, we revised this part as follows:

Lines 136-138: “possible mass fluxes driven by other mechanisms (e.g. thermal convection, viscous convection and concentration gradient).…”

6. “IMIC. The model by Ishii and Dziewoński (2002) is based on ISC travel times (not so accurate) and a bit outdated. There are various models with updated results of IMIC radius and anisotropy pattern (e.g., Sun and Song 2008 EPSL; Ref. 48 Wang et al. 2015 Ngeo; Stephenson et al., 2020 JGR). Can your model fit well with more recent results?”

[Authors]: Thank you for your suggestion. We compared with the data of Ishii and Dziewoński 2002 because the data are absolute PKIKP velocities without other reference phases. According to your suggestion, we also compared the data Sun and Song 2008 in the revised Figure. 4c. The observations of Sun and Song 2008 present consistent results, showing the slowest axis approximately 50 degrees from Earth’s rotation axis. Thus, the AEP model fits better with these data. For the references of Wang et al. 2015 Ngeo and Stephenson et al., 2020 JGR, we did not find the absolute or relative (compared with ak135) velocities in their publications. Thus, it is difficult

for us to process their data and compare them in this study. We also added the following discussion in the revision:

Lines 165-167: “In the IMIC (green region), the seismic anisotropy presents the slowest angle of $\sim 45\text{-}50^\circ$, and the AEP model fits better with the absolute travel-time data¹³ and PKPab-PKIKP data for IMIC³⁸ (Fig. 4c).”

Fig. 4 | Schematic diagram of the IC geomagnetic field and depth-dependent anisotropic texture change in comparison with the calculated velocity anisotropy in FeH_{0.25}. (a) Poloidal (BP) and toroidal (BT) fields in the IC in the meridional plane. BP is shown with red curves in the right sphere, and BT is shown with blue curves in the left sphere. The dashed purple line is the rotation axis. The black line is the PKIKP ray path. The angle between the ray path and Earth's rotation axis is ξ . The isotropic layer at the UIC is shown with a yellow region. The deeper layer presenting the slowest velocity in the equatorial direction is shown with the blue region. The IMIC with the slowest angle of $\sim 45\text{-}50^\circ$ from Earth's rotation axis is shown with a green region. (b) PKPab-PKIKP travel-time residuals¹⁵ as a function of angle ξ are compared with the calculated compressional wave velocity anisotropies in the AEP (black curves) and IEP (blue curves) models. PKIKP refers to compressional waves passing through the inner core. PKPab is the reference seismic wave phase traversing only the mantle and outer core. The data travelling from the South Sandwich Islands (SSI) to Alaska are shown with red circles. (c) Calculated velocity anisotropy in FeH_{0.25} in comparison with the velocities of PREM⁵² (1221 km, red dashed line) and the anisotropic IMIC (920-1221 km, green and orange dashed curves) modes^{13,51}.

7. "A more interesting question is: Can your model explain the equatorial anisotropy in the IMIC, such as proposed by Wang et al. 2015? The key observation was a symmetry axis nearly parallel to the equatorial plane, rather than the change of the form of the anisotropy and how the anisotropy changes with ray angles."

[Authors]: It is very a very good question. We have tried to construct a model to explain the equatorial anisotropy. Interestingly, a circular alignment of the lattice is able to explain the observation of Wang et al. (2015). However, other patterns with many circular textures can also explain the equatorial anisotropy. The radius of this circle should be related to the strength of the magnetic field in the IC. In this case, we still need to calculate the electric field in the IC to understand such a complex structure, and we agree with the reviewer that equatorial anisotropy is a very important and interesting project for our future study.

8. “Anisotropy complexities. The authors also attempt at some IC anisotropy complexities, such as hemispherical dichotomy (L169-171). However, it is qualitative (which is fine at this stage), but the explanation for the hemispherical variation is not convincing. The “hemispherical” pattern is not exactly hemispherical (Sun and Song 2008 PEPI) either. In general, the anisotropy is quite complex, is fully 3D. On the other hand, it’s likely that the geomagnetic field and electric current in the IC are also likely complex. That seems much more convincing cause for the anisotropy complexities. With field from geodynamo simulations, quantitative arguments can certainly be worked out.”

[Authors]: Thank you for this valuable suggestion. We agree with the reviewer that geodynamo simulations are needed for a quantitative explanation of hemispherical dichotomy. Accordingly, we revised this part as follows:

Lines 173-180: “This mechanism establishes a connection between the inner core structure and electric field structure of the IC. It is most likely that the electromagnetic field in the IC is complex, as the dipole geomagnetic field is tilted and eccentric to Earth’s rotational axis. Based on reconstructions of the magnetic field from the past 10,000 years, the geomagnetic field in the core can be described as eccentric to the west by approximately 100 km^{48,49}. It may also suggest an eccentric electric field in the IC, and specific calculations on the electric field in the IC considering the characteristics of eccentricity and tilting are necessary. This may provide a clue to understand the hemispherical variations in velocity anisotropy and the complex inner core structure.”

8. “Time varying anisotropy? Magnetic field certainly changes with time. As it changes, will the IC anisotropy changes too? How long does it take? Temporal changes of the IC have been well observed (Song and Richards 1996 Nature ...). Would the anisotropy

change be observable in human time scale?”

[Authors]: It is an important question. We try to make a rough estimation on the time scale. The strength and structure of the magnetic field in the outer core is poorly constrained. The estimated geomagnetic field strength based on tidal dissipation is approximately 2.5 mT (Buffett 2010 Nature). In this case, the current density is estimated to be 1 mA m⁻² at the ICB. The conductivity of the Fe-H alloy was on the order of 10⁶ S m⁻¹, which was calculated in our previous study (He et al., 2022 Nature). Here, we also assume that the recrystallization of the Fe-H alloy at inner core temperatures is much faster than the process of magnetic diffusion and the accumulation of diffusion-induced stress. This assumption is reasonable, as a previous study has shown that the recrystallization of alloys is quite fast (tens of hours) when the temperature is close to the melting temperature (Bergman et al., 2010 GRL). Geodynamo simulations suggest that it takes thousands of years for the diffusion of a magnetic field (B_P) into the deepest part of the IC. In this case, the change in seismic anisotropy in the deepest part of the IC may take thousands of years. For the shallow part, the time scale may depend on the process of stress accumulation. Based on our simulation in this study, it may take ~3-30 years for the accumulation process, and stress may promote the alignment of the lattice.

9. “Anisotropy models (IEP and AEP) (again). The authors consider that the texture alignment of the Fe-H alloy is driven by the geomagnetic field. The angle ξ defined in this study is based on the Earth's rotation axis, but the geomagnetic pole is not the same as the rotation axis. This should be explained or at least pointed out in the manuscript or supplement.”

[Authors]: Thank you for this valuable suggestion. We explained this in the revised main text:

Line 134-135: “...It is worth noting that we did not consider the possible tilting of B_P .”

Minor comments:

L57. “these alloys” -> clarify. “these alloys (with the light elements)”?

[Authors]: Thank you. It was changed to superionic Fe alloys.

L62. “anisotropic compressional velocity (AVp)”. This is confusing. Do you mean “compressional velocity anisotropy”?

[Authors]: Thank you. We revised it to compressional velocity anisotropy.

L105-108. “Generally, we have .. ”. This seems a summary sentence of the section. You probably should put it in a separate paragraph.

[Authors]: Thank you. We started a new paragraph and added more discussion in this part.

L391 (Figure 1b), the thick dashed gray line connects the troughs of the top three curves, but not including the bottom one. A mistake or on purpose? It looks a bit weird.

[Authors]: Thank you. The thick gray line is the change trend of the slowest axis with respect to hydrogen content. This was noted in the revised caption.

L420, Figure 4. You may illustrate the definition of ξ in (a), instead of in the lengthy caption, by adding a raypath and labeling the angle in a suitable place.

[Authors]: Thank you for the kind suggestion. We revised Fig. 4a accordingly.

In plots of dt/t , such as Figure 4b, tell how the curves are calculated. At what distance/sampling depth?”

[Authors]: Thank you for the important question. We calculated dt/t by $\frac{\delta t}{t} = \frac{V_P - V_{P0}}{V_{P0}}$,

where V_P is the velocity change along with ξ . It was calculated using elastic constants at 360 GPa and 6000 K (~821 km below ICB). We also calculated the V_P using the data calculated at 330 GPa and 5500 K (~21 km below ICB), and the influence of pressure on the anisotropy is negligible. V_{P0} is calculated based on the estimated velocities in

polycrystals within the Voigt scheme (He et al., 2021 PRB). This method was used in previous velocity anisotropy calculations on pure iron (Mattesini et al., 2010 PNAS; Stixrude & Cohen, 1995 Science).

Response to the anonymous Reviewer #3:

Sun et al., conducted theoretical calculations on elastic properties of the superionic Fe-H alloys at the inner core relevant P-T as well as electric field conditions. Their results showed that the geomagnetic field of inner core will align the c-axis of superionic Fe-H parallel to the equatorial plane in deeper depth instead of the uppermost region (both B_p and B_t exist). This can well explain the observed depth-dependent anisotropy in the inner core. Although the reason for the hemispheric dichotomy is still unclear, this study provides a new aspect to understand the possible coupling and interaction between the inner core materials with the Earth's geomagnetic field. There are some questions after reading the manuscript.

[Authors]: Thank you for your comments and suggestions. We revised accordingly.

The authors calculated the AVp ranging 3.9-5.3% up to 6000 K, which is the maximum anisotropy of the hcp superionic Fe-H alloy. From Figures 1. S6 and S8, I found that the velocity difference between c- and a-axis is relatively small, about 1.7% at 6000 K and almost 0% at 4000 K. Under geomagnetic field, the c-axis of superionic Fe-H is aligned to be perpendicular to the Earth's rotation axis and the a-b plane is parallel. I wonder whether the velocity difference in c and a-axis can well explain the seismically-observed anisotropy magnitude in the inner core.

[Authors]: Thank you for your careful reading and the important question. We used the data of FeH_{0.25} at 360 GPa and 6000 K to compare with seismological observations, as shown in Figure 4b and c. Here, we considered two conditions for the alignment of the c-axis in the equatorial plane. In the IEP model, the c-axis is randomly distributed, and the velocity change from the rotation axis to the equatorial plane equals the average of

the velocity changes from the fast axis (c-axis) to all possible azimuthal angles, from the c-axis to the a-axis. For the anisotropic model, the alignment of the c-axis may form some anisotropic patterns. Here, we considered the most extreme condition assuming that the pattern is able to allow the velocity variation with ξ to be consistent with the velocity change from the a-axis to the c-axis. In Fig. 4b, the IEP model can fit with the observed anisotropy at depths less than 1100 km. Based on our estimation, approximately 65%-80% of aligned FeH_{0.25} is able to explain the observed seismic anisotropy. In the inner most inner core (IMIC), the slowest axis is approximately 50 degrees from the rotation axis. Thus, the AEP model fit with the observations better. As we mentioned in the manuscript, this model is still simple and cannot exactly match the seismological observations, and further studies on the effect of other light elements and/or inhomogeneous distribution of light elements are needed.

The authors found the the velocity of superionic Fe-H in c-axis becomes smaller than that in a-axis when temperature increasing to 5000 and 6000 K (so-called reversal of fastest direction). Such a phenomenon is not observed by Wang et al., calculating Fe₆₄H₄ at 6500 K and 360 GPa. What is the reason? In He et al., 2022, they suggested that Fe-O and Fe-C alloys can also be superionic phases at inner core relevant P-T. I wonder if the C and O would change the story or the observation here.

[Authors]: Thank you for the important question. As shown in Figure 1 and Figure S4, the reversal of the fastest direction is due to the increase in the c/a ratio with different temperatures and H contents. The hydrogen content in the study by Wang et al. is lower (Fe₆₄H₄), which may be the reason that the effect was not observed. For superionic FeC_{0.0625} and FeO_{0.0625}, the fastest axis is still along the c direction. In this case, hydrogen is critical for this model. Recent studies also suggest strong partitioning of hydrogen into liquid Fe over silicate melts, which suggests high hydrogen content in Earth's core (Li et al., 2020 Nat. Geosci.; Yuan & Steinle-Neumann, 2020 GRL; Tagawa et al., 2021 Nat. Commun.).

Based on Figure 4b, the anistropy in the inner core can be roughly divided to three

layers, the uppermost (almost no anisotropy), the middle (some anisotropy), and the innermost (strong anisotropy) regions. It is conceivable that the existence of both Bp and Bt in the uppermost region make it almost no anisotropy. However, it is unclear to me why there is a great increase in anisotropy in the innermost region from the manuscript. At lines 150-151, the authors mention “This may suggest the presence of a highly anisotropic equatorial plane in the IMIC”. I wonder what is such a plane and why it can exist? Can it be related to the properties of superionic Fe-H? The depth starting from? Please clarify or elaborate.

[Authors]: It is a very interesting question. The anisotropy change in the IMIC is suggested by seismological observations. In the IMIC, the slowest direction changes from the equatorial plane to a direction ~ 50 degrees from the rotation axis. The AEP model also presents the slowest axis at $\sim 50^\circ$, which is consistent with seismic data (Ishii & Dziewonski 2002 PNAS & Sun & Song 2008 EPSL). This suggests possible anisotropic equatorial anisotropy. The equatorial anisotropy is also suggested by a recent seismology study using the coda-correlation wavefield (Wang et al., 2015 Nat. Geosci.). We have tried to construct the anisotropic pattern in the equatorial plane. A circular alignment of the lattice is able to explain the observations by Wang et al. 2015. However, this is a multiple solutions issue without the support of electric field distribution in the IC. The formation mechanism is also a very important question. This may be due to electric field changes at the IMIC, which might be a consequence of the past geomagnetic field.

Although the existence of the superionic Fe-H can explain the anisotropy with depth to some extent, it is still unclear how it causes the hemispheric dichotomy. Some efforts or attempts may be worthwhile on this part.

[Authors]: Thank you for this valuable suggestion. Indeed, the inner core structure is complex and not easy to understand without information on the electric field distribution in the inner core. As the dipole axis of the geomagnetic field is eccentric and tilted with respect to Earth's rotation axis, the magnetic field in Earth's inner core should also be eccentric and tilted. In this case, it is easy to understand the reason for

the IC complex structure. For hemispheric dichotomy, geodynamo simulations and palaeomagnetic observations support the offset of the geomagnetic center to the west site in recent 10000 years. This study provides an important clue to understanding the hemispheric dichotomy, and more studies, especially geodynamo simulations, are needed for comparison with seismic data. Based on your suggestion, we improved the description of this part as follows:

Lines 173-180: “This mechanism establishes a connection between the inner core structure and electric field structure of the IC. It is most likely that the electromagnetic field in the IC is complex, as the dipole geomagnetic field is tilted and eccentric to Earth’s rotational axis. Based on reconstructions of the magnetic field from the past 10,000 years, the geomagnetic field in the core can be described as eccentric to the west by approximately 100 km^{48,49}. It may also suggest an eccentric electric field in the IC, and specific calculations on the electric field in the IC considering the characteristics of eccentricity and tilting are necessary. This may provide a clue to understand the hemispherical variations in velocity anisotropy and the complex IC.”

Minor comments:

In Figure 4, the caption should be “c”. There is no “d”. In Methods, the AVs is actually the maximum shear wave splitting anisotropy instead of the shear wave anisotropy.

[Authors]: Thank you for pointing out the mistakes. We revised accordingly.

In figure S2, P, T, and H contents are three variables that change. It is unclear from the caption the exact fixed values in each category.

[Authors]: Thank you for the good suggestion. We revised the caption accordingly.

In tables S1 and S2, the c_{ij} has a unit of GPa?

[Authors]: Thank you for pointing out the mistakes. We revised accordingly.

REVIEWER COMMENTS

Reviewer #1 (Remarks to the Author):

The authors have addressed my criticisms, I believe the paper is now publishable.

Reviewer #2 (Remarks to the Author):

Review of the revised ms
Xiaodong Song, Peking U

From the revised ms and the rebuttal letter, I think the authors have done a very good job in addressing the reviewers' concerns. They have addressed my main concerns. I recommend publication, after addressing some minor questions below.

--Line 184. I didn't see how the study fit the seismic density. This is also mentioned in the rebuttal letter. Please elaborate. Although the main interest is anisotropy, the reasonable fits to absolute values of V_p , V_s and density do make the model more appealing.

--Supp. Fig. 7. The descriptions and the illustrations of the IEP and AEP are still confusing. Panel a. perhaps, label $c(eq)$ and $a(ns)$. How does a circular arrow represent an axis? Get rid of it; label N, S instead.

Panel b. what's the aligned pattern on the left? C axis is in circular direction? Is basal plane also parallel to the rotation axis (like in a)?

--In the rebuttal letter, the authors stated:

"Interestingly, a circular alignment of the lattice is able to explain the observation of Wang et al. (2015)."

This is intriguing. Wang et al. 2015 proposes equatorial anisotropy with a fast axis nearly parallel to the equatorial plane. How does your model explain it?

Reviewer #3 (Remarks to the Author):

Thanks for the authors' efforts in revising the manuscript.

It is interesting that the authors constructed the IPE model, that can well fit the seismic observation (Figure 4b). But I do not quite understand how this calculation was made. It is stated that in the IPE model, the c-axis is randomly distributed. It is a hcp structure. If the c-axis is randomly distributed, I would expect the crystals are randomly distributed. Does that mean there will be no anisotropy? The authors stated that "In this model, the change in the velocity as a function of angle ξ between the ray path and Earth's rotation axis is calculated by averaging the velocities of all the possible paths from the fast axis to the a-c plane in the hcp structure." Could you please add more justification and clarification on this? Adding some schematic figures will help readers to understand. The current Figure S7 is not sufficient.

I got the explanation why the reversal of fastest direction not observed by Wang et al., 2021. The authors may need to add sentences in main text or SI for discussion. Indeed, the H content may play a role. Since there is lacking the FeO_{0.25} or FeC_{0.25} data, the elements may also play a role. Such information is important for readers.

Response Letter

Reviewer #1 (Remarks to the Author):

The authors have addressed my criticisms, I believe the paper is now publishable.

[Authors]: Thank you for the suggestion for publication.

Reviewer #2 (Remarks to the Author):

Review of the revised ms Xiaodong Song, Peking U

From the revised ms and the rebuttal letter, I think the authors have done a very good job in addressing the reviewers' concerns. They have addressed my main concerns. I recommend publication, after addressing some minor questions below.

[Authors]: Thank you for the suggestion for publication. We revised according to your comments.

--Line 184. I didn't see how the study fit the seismic density. This is also mentioned in the rebuttal letter. Please elaborate. Although the main interest is anisotropy, the reasonable fits to absolute values of V_p , V_s and density do make the model more appealing.

[Authors]: Thank you for the question. We have shown calculated V_p , V_s and densities of Fe-H alloys at different pressures and temperatures. The data is compared with the PREM data under inner core conditions as shown in Supplementary Fig. 2. The calculated V_s is still larger than that of the PREM data. As suggested by previous studies (Lin et al., 2003 GRL & Belonoshko, et al., 2007 Science), V_s may be further reduced due to the presence of nickel and the effect of grain boundaries.

Supplementary Fig. 2 | Calculated seismic velocities in Fe-H alloys in comparison with the PREM data. (a) Compressional wave velocities (V_p) and (b) shear wave velocities (V_s) in FeH_{0.25} at temperatures from 0 to 6000 K and 360 GPa are shown

with green symbols. The V_P and V_S in $\text{FeH}_{0.25}$ at 5500 K and 330 GPa are shown with blue symbols. V_P and V_S in $\text{FeH}_{0.0625}$ and $\text{FeH}_{0.125}$ at 360 GPa and 6000 K are shown with orange symbols. The trends of increasing temperature, pressure and H content are shown with light green, light blue, and yellow arrows.

--Supp. Fig. 7. The descriptions and the illustrations of the IEP and AEP are still confusing. Panel a. perhaps, label c(eq) and a(ns). How does a circular arrow represent an axis? Get rid of it; label N, S instead.

Panel b. what's the aligned pattern on the left? C axis is in circular direction? Is basal plane also parallel to the rotation axis (like in a)?

[Authors]: Thank you for the suggestions. We revised Supp. Fig. 7 accordingly. Yes, the c axis is in circular direction and the basal plane parallel to the rotation axis. We also revised the aligned pattern by adding the direction of c axis in the pattern.

--In the rebuttal letter, the authors stated: "Interestingly, a circular alignment of the lattice is able to explain the observation of Wang et al. (2015)."

This is intriguing. Wang et al. 2015 proposes equatorial anisotropy with a fast axis nearly parallel to the equatorial plane. How does your model explain it?

[Authors]: It is a very important question. Indeed, the AEP model make it possible to construct anisotropic seismic structure in equatorial plane. As shown in the following figure, we construct a circular alignment equatorial plane model. The c-axis is in the circular direction. Here, the center of circular (black star) presents some distance from the center of the inner core. This simple model can generate anisotropic seismic velocities in the equatorial plane. For the blue axis, it is perpendicular to c axes of hcp Fe-H crystals presenting as a fast axis in this equatorial model. This fast axis is able to explain the observation of Wang et al. 2015 as the ray paths of PKIKP² pass the center of the inner core. However, other patterns with the alignment of c-axis perpendicular to a ray path may also lead to a fast axis in equatorial plane. In this case, calculation of electric field in the IC is important to construct a reliable anisotropic equatorial model.

Reviewer #3 (Remarks to the Author):

Thanks for the authors' efforts in revising the manuscript.

It is interesting that the authors constructed the IPE model, that can well fit the seismic observation (Figure 4b). But I do not quite understand how this calculation was made. It is stated that in the IPE model, the c-axis is randomly distributed. It is a hcp structure. If the c-axis is randomly distributed, I would expect the crystals are randomly distributed. Does that mean there will be no anisotropy?

[Authors]: Thank you for your good question. As mentioned in the line 145-147 of the manuscript, “In the isotropic equatorial plane (IEP) model, the alignment of the c-axis in the equatorial plane is randomly distributed ...” We mean that c-axis is aligned perpendicularly to the polar direction and distributed randomly in the equatorial plane, so the velocity is isotropic in the equatorial plane and behaves anisotropy as the ray-angle (ξ) changes between the equatorial plane and polar direction. To make this IEP model more comprehensible, we have improved our expressions as following:
Line 145-147: “In the isotropic equatorial plane (IEP) model, c-axis of hcp Fe-H crystals are aligned perpendicularly to the polar direction and randomly distributed in the equatorial plane.”

The authors stated that “In this model, the change in the velocity as a function of angle ξ between the ray path and Earth’s rotation axis is calculated by averaging the velocities of all the possible paths from the fast axis to the a-c plane in the hcp structure.” Could you please add more justification and clarification on this? Adding some schematic figures will help readers to understand. The current Figure S7 is not sufficient.

[Authors]: Thank you for your constructive question and precious suggestion. To better explain the IEP model and give detailed process of calculation, we plotted a schematic figure into supplementary Fig.7 as following:

Hcp Fe-H crystal is simplified as a cylinder in considering of its cylindric symmetry. In the IEP model, c-axis is assumed to be perpendicular to the polar direction. Velocity in (001) plane (a-b plane) is equivalent in symmetry. We can arbitrarily choose a direction in (001) plane as the axis along the polar direction and set the plane perpendicular to it be the equatorial plane. For example, we can let a-axis ([100] direction of crystal) aligned along the polar direction and set the (100) plane of crystal in the equatorial plane. The propagation direction vector is $\mathbf{n}(n_1, n_2, n_3)$ as mentioned in SI equation (14). The crystal has a rotational degree of freedom θ in the equatorial plane. For a given ray-angle (ξ), θ denote the angle between the projection line (red dashed line) of ray in equatorial plane and c-axis. Thus, we have:

$$\begin{cases} n_1 = \cos \xi \\ n_2 = \sin \xi \sin \theta \\ n_3 = \sin \xi \cos \theta \end{cases} \quad (1)$$

and we can get $V_P(\xi, \theta)$ by substituting $\mathbf{n}(\cos \xi, \sin \xi \sin \theta, \sin \xi \cos \theta)$ into supplementary equation (14) and solving the Christoffel equation. In the IEP model, we set the sampling interval of θ as 1° and $V_P(\xi)$ was calculated by:

$$V_P(\xi) = \frac{1}{360} \sum_{\theta} V_P(\xi, \theta), \theta = 1, 2, \dots, 360 \quad (2)$$

These discussions were added to SI. We have also improved the sentence as following:
Line 225-228: “In this model, the change in the velocity as a function of angle ξ between the ray path and Earth’s rotation axis is calculated by averaging the velocities

of all the possible propagation direction paths corresponding to the different alignment of crystal with a rotatable c-axis in the equatorial plane (Supplementary Fig. 7).”

We replotted Supplementary Fig. 7 as following:

Supplementary Fig. 7 | Anisotropic Models for hcp-FeH_{0.25} with the basal plane along Earth's rotation axis. a Isotropic equatorial model (IEP) model with c-axis aligned perpendicularly to the polar direction and randomly distributed in equatorial plane (left and middle panel). In the right panel, the red dashed line in equatorial plane denotes the projection line of the ray path. For a given ray-angle (ξ), the crystal has a rotational degree of freedom θ in the equatorial plane. This model is viewed as a cylindrically averaged (averaged over θ) aggregate with a basal plane parallel to the rotation axis. **b** c-axis is aligned in some patterns (left and middle panel) to allow the velocity variation with ξ is consistent with the velocity change from the basal plane to

the c-axis (right panel).

I got the explanation why the reversal of fastest direction not observed by Wang et al., 2021. The authors may need to add sentences in main text or SI for discussion. Indeed, the H content may play a role. Since there is lacking the FeO_{0.25} or FeC_{0.25} data, the elements may also play a role. Such information is important for readers.

[Authors]: Thank you for your valuable suggestion. We added following sentences in the main text to explain the difference. “Beyond the temperature effect, we also evaluated velocity anisotropy in Fe-H alloy with different hydrogen content (Fig. 1b). FeH_{0.0625} still presents the fastest direction along c-axis like pure hcp-Fe, which is consistent with pervious study⁴⁴. Further increasing the hydrogen content leads to a reversal of the fastest direction.”

Based on the experimental studies, the solubility of C in hcp-Fe is about 1 wt. % at ~211 GPa (Mashino et al., EPSL 2019), while the O concentration in solid Fe can be negligible (Ozawa et al., PEPI 2010). In this case, further increasing C and O content may make the system unstable and unrealistic under inner core conditions.

REVIEWERS' COMMENTS

Reviewer #3 (Remarks to the Author):

Thank the authors for their great efforts in revising the manuscript to address the relevant questions.
I do not have further questions at the current stage.
I do think it deserves an acceptance for publication.

Response to Reviewer

Reviewer #3 (Remarks to the Author):

Thank the authors for their great efforts in revising the manuscript to address the relevant questions. I do not have further questions at the current stage.

I do think it deserves an acceptance for publication.

[Authors]: Thank you for the valuable suggestions for the improvement of our manuscript, and the suggestion for publication.